# Detecting the Potential for Consciousness in Unresponsive Patients Using the Perturbational Complexity Index

**DOI:** 10.3390/brainsci10120917

**Published:** 2020-11-27

**Authors:** Dmitry O. Sinitsyn, Alexandra G. Poydasheva, Ilya S. Bakulin, Liudmila A. Legostaeva, Elizaveta G. Iazeva, Dmitry V. Sergeev, Anastasia N. Sergeeva, Elena I. Kremneva, Sofya N. Morozova, Dmitry Yu. Lagoda, Silvia Casarotto, Angela Comanducci, Yulia V. Ryabinkina, Natalia A. Suponeva, Michael A. Piradov

**Affiliations:** 1Research Center of Neurology, Volokolamskoe Shosse, 80, Moscow 125367, Russia; alexandra.poydasheva@gmail.com (A.G.P.); bakulinilya@gmail.com (I.S.B.); milalegostaeva@gmail.com (L.A.L.); lizaveta.mochalova@gmail.com (E.G.I.); dmsergeev@yandex.ru (D.V.S.); lavrentevan@mail.ru (A.N.S.); Moomin10j@mail.ru (E.I.K.); kulikovasn@gmail.com (S.N.M.); dmitrylagoda.doc@gmail.com (D.Y.L.); Ryabinkina11@mail.ru (Y.V.R.); nasu2709@mail.ru (N.A.S.); mpi711@gmail.com (M.A.P.); 2Department of Biomedical and Clinical Sciences “L. Sacco”, University of Milan, 20157 Milan, Italy; silvia.casarotto@unimi.it; 3IRCCS Fondazione Don Carlo Gnocchi ONLUS, 20148 Milan, Italy; acomanducci@dongnocchi.it

**Keywords:** disorders of consciousness, unresponsive wakefulness syndrome, minimally conscious state, diagnosis, perturbational complexity index, electroencephalography, transcranial magnetic stimulation, reliability

## Abstract

The difficulties of behavioral evaluation of prolonged disorders of consciousness (DOC) motivate the development of brain-based diagnostic approaches. The perturbational complexity index (PCI), which measures the complexity of electroencephalographic (EEG) responses to transcranial magnetic stimulation (TMS), showed a remarkable sensitivity in detecting minimal signs of consciousness in previous studies. Here, we tested the reliability of PCI in an independently collected sample of 24 severely brain-injured patients, including 11 unresponsive wakefulness syndrome (UWS), 12 minimally conscious state (MCS) patients, and 1 emergence from MCS patient. We found that the individual maximum PCI value across stimulation sites fell within the consciousness range (i.e., was higher than PCI*, which is an empirical cutoff previously validated on a benchmark population) in 11 MCS patients, yielding a sensitivity of 92% that surpassed qualitative evaluation of resting EEG. Most UWS patients (*n* = 7, 64%) showed a slow and stereotypical TMS-EEG response, associated with low-complexity PCI values (i.e., ≤PCI*). Four UWS patients (36%) provided high-complexity PCI values, which might suggest a covert capacity for consciousness. In conclusion, this study successfully replicated the performance of PCI in discriminating between UWS and MCS patients, further motivating the application of TMS-EEG in the workflow of DOC evaluation.

## 1. Introduction

Prolonged disorders of consciousness (DOC) represent a spectrum of syndromes occurring in patients who survive coma after severe brain injury. Their characteristic feature is the dissociation between preserved arousal and complete or partial absence of awareness [1]. The clinical characterization of DOC is important for prognosis and for personalizing the rehabilitation treatment. Indeed, DOC severity has been related to the likelihood of recovery of consciousness [2,3,4] and rehabilitation interventions with neuromodulatory approaches have shown greater efficacy in MCS patients [5,6,7,8,9,10]. However, differential diagnosis between patients who show reflexive movements only (unresponsive wakefulness syndrome—UWS) and patients able to perform inconsistent albeit reliable purposeful behaviors (minimally conscious state—MCS) is often challenging [11,12].

Therefore, the search for methods capable of accurately detecting the presence of consciousness in unresponsive patients is one of the major goals in DOC research. Along these lines, the integration of the gold-standard behavioral assessment—based on the Coma Recovery Scale Revised (CRS-R) [13]—with a multimodal quantitative neurophysiological and functional neuroimaging approach has been repeatedly suggested by the most recent international guidelines and reviews on DOC [2,14,15,16]. In particular, selected electrophysiological studies were recommended whenever ambiguity regarding the presence of consciousness in unresponsive patients emerges from behavioral examination. Among these suggested complementary evaluations, the perturbational complexity index (PCI), which is obtained from the electroencephalographic (EEG) responses to transcranial magnetic stimulation (TMS) [17], represents one of the most promising tools for objectively evaluating the state of consciousness in severely brain-injured patients. Indeed, validation of PCI in a large benchmark population, including healthy controls as well as communicative brain-injured subjects, provided an empirical cutoff (PCI* = 0.31) able to perfectly discriminate between unconsciousness (non-rapid eye movement sleep, anesthesia with propofol, midazolam and xenon) and consciousness (wakefulness), even if disconnected (dreaming and ketamine anesthesia) [18]. This externally validated, empirical cutoff was then applied to 81 DOC patients: results showed that PCI (i) was able to detect with unprecedented sensitivity (94.7%) MCS patients and (ii) allowed to identify a subgroup of UWS patients with a potentiality for consciousness not expressed in behavior.

These important findings led to the inclusion of PCI in the most recent guidelines for DOC diagnosis: in particular, this index—which is based on TMS-EEG—has been suggested to distinguish MCS from UWS patients to a mildly important degree [2]. Furthermore, TMS-EEG has also been proposed as part of multimodal assessment for its high sensitivity and specificity in differentiating between UWS and MCS patients [14].

To further increase the evidence and strengthen the level of recommendation in future guidelines, it is essential that different research groups may confirm the promising results of PCI by performing studies that match the strict methodological criteria requested by a systematic review (e.g., including DOC patients classified according the clinical reference standard, providing data which allows calculation of diagnostic accuracy or objectively testing patients without knowledge of the patient’s clinical status). Indeed, the prospects of adopting TMS-EEG as an advanced diagnostic tool in the clinical practice would benefit from a confirmation about the accuracy of this technique in heterogeneous clinical contexts characterized by different clinical populations, equipment, and expert judgement in the data acquisition and analysis.

In order to promote the application of PCI in future research and clinical settings, in the present study we tested the reliability of PCI as a diagnostic marker by applying the same experimental and computational procedure as well as the same empirical cutoff reported in [18] in prolonged (at least 28 days from the brain injury [2,19]) and chronic (at least 3 months in non-traumatic and 12 months in traumatic cases [2,19]) DOC patients enrolled at the Research Center of Neurology (Moscow, Russia).

## 2. Materials and Methods

### 2.1. Patients

Twenty-four severely brain-injured patients participated in this study (10 female; age range 19–55 years; time since injury between 2–56 months; etiology: traumatic *n* = 12, vascular *n* = 4, anoxic *n* = 6, other *n* = 2; for more details see Table 1). Informed consent to participation was obtained from the patients’ legal representatives. Upon enrollment, all patients were screened for potential adverse effects of TMS and were in a stable clinical condition for at least two weeks. The patients did not receive any medication affecting the nervous system (including antiepileptic drugs) at the time of and at least seven days prior to the EEG and TMS-EEG data collection and CRS-R assessment.

The study was approved by the Ethical Committee of the Research Center of Neurology (Moscow, Russia), protocol 11/14, date of approval: 19 November 2014.

### 2.2. Experimental Protocol

In each patient, the state of consciousness was clinically evaluated following the validated Russian version [20] of the Coma Recovery Scale Revised (CRS-R) [13] by one of three trained neurologists (L.A.L., E.G.I., D.V.S.) The scale consists of 23 items comprising 6 subscales that assess auditory, visual, motor, oromotor, communication, and arousal processes. Each subscale score is determined by the presence of specific behavioral responses to sensory stimuli administered in a standardized manner, the lowest item representing reflexive activity and the highest one corresponding to cognitively mediated behaviors. The diagnosis of minimally conscious state (MCS) or unresponsive wakefulness syndrome (UWS) was set according to the best diagnosis (Table 1) observed across five CRS-R assessments performed over a period of 2–3 successive days. MCS patients were further classified as MCS+ and MCS− according to the criteria recently proposed by [21].

Within a week from the clinical evaluation, each patient underwent a neurophysiological assessment including resting-state EEG and TMS-EEG data collection. The entire experimental procedure, including EEG montage, lasted approximately 2.5–3 h. During the recordings, the researcher always ascertained that patients kept their eyes open either spontaneously or after a behavioral stimulation [13], in order to keep a stable arousal state. For TMS neuronavigation, a T1-weighted anatomical magnetic resonance image (MRI) was also acquired (Siemens Verio 3T, GR/IR sequence, TE/TR = 2.47/1900 ms, flip angle 9°, isotropic voxel size 1.0 mm^3^).

The present study employed the same TMS-EEG equipment and followed similar experimental procedures as described in [18]. Specifically, resting and TMS-evoked EEG data were collected with a 60-channel TMS-compatible EEG amplifier (Nexstim Ltd., Helsinki, Finland). The duration of resting EEG recordings was at least of 10 min.

A physical reference electrode was located on the forehead. An electrooculogram (EOG) was bipolarly recorded using two additional electrodes with a diagonal montage. Impedance at all electrodes was kept below 5 kΩ. Raw data were band-pass filtered between 0.1 and 350 Hz and sampled at 1450 Hz with 16-bit resolution. During TMS stimulation, auditory potentials elicited by TMS-associated click sound were maximally reduced by continuously playing a masking noise through inserted earplugs. TMS was delivered with a Focal Bipulse 8-Coil (mean/outer winding diameter ca. 50/70 mm, biphasic pulse shape, pulse length ca. 280 ms, focal area of the stimulation hot spot 0.68 cm^2^; eXimia TMS Stimulator, Nexstim Ltd., Helsinki, Finland). A Navigated Brain Stimulation (NBS) system (Nexstim Ltd., Helsinki, Finland) was used to set stimulation parameters on individual MRIs and to ensure their stability across trials in real-time. The coil was always placed tangentially to the scalp to maximize the impact of TMS on the cortex. Similar to [18], TMS pulses were mainly delivered to the superior frontal and/or parietal cortex, approximately 1 cm lateral to the midline, with an inter-pulse interval randomly jittered between 2 and 2.3 s. Recharging of the stimulator’s capacitors was delayed to 1000 ms to prevent the occurrence of an artifact within the temporal window of interest for TMS-evoked brain activity. Since the direct stimulation of cortical lesions does not elicit any measurable EEG response [22], TMS targets were selected over structurally preserved areas, far away from cortical lesions, to maximize the possibility of obtaining complex responses. Stimulation intensity was set so as to induce an electric field of at least 120 V/m on the cortical surface, as estimated by the NBS system. In addition, EEG responses to TMS were visually monitored in real time to minimize muscle artifacts and, at the same time, to maximize the impact of TMS on the cortex, by slightly adjusting the position and orientation of the TMS coil.

### 2.3. Data Analysis

TMS-EEG data analysis was performed as described in [17,18] by two researchers (D.O.S. and S.C.) who were blind to the behavioral diagnostic category of DOC patients. Artifact-contaminated single trials and channels were rejected by visual inspection. Raw data were high-pass filtered at 1 Hz and split into symmetric 1.6-second-long epochs around the TMS pulse. Then, Independent Component Analysis was applied to reduce ocular and muscle artifacts. Notch filtering at 50 Hz and low-pass filtering at 45 Hz were applied to the signal reconstructed after visual rejection of the artifact-contaminated independent components. Finally, pre-processed data were shortened to ±400 ms around the pulse, downsampled to 362.5 Hz, and used for estimating cortical current density distribution over time. As described in [17], a bootstrap-based statistical analysis was applied to the spatio-temporal distribution of cortical currents for extracting the deterministic pattern of TMS-evoked cortical activity at the source level. The Lempel–Ziv complexity of the binary spatio-temporal matrix resulting from statistical analysis between 8 and 300 ms, normalized by source entropy, was computed to derive the PCI values. Whenever source entropy was ≤0.08, corresponding to ≥1% probability of type I error, PCI value was set to 0, which indicated the absence of a significant TMS-evoked activation. For each patient, the maximum value of PCI (PCI_max_) across stimulation sites was considered for diagnostic classification and compared with the previously published empirical cutoff PCI* = 0.31, validated in a large benchmark population to discriminate between unconsciousness and consciousness [18].

Resting-state EEG data were pre-processed as described in [18] and analyzed by a trained neurophysiologist (A.C.) who was blind to behavioral diagnosis. Specifically, continuous EEG recordings were re-referenced to the standard double banana montage, band-pass filtered between 1–70 Hz and downsampled to 725 Hz. Then, qualitative assessment was performed according to the following EEG descriptors: predominant background frequency, organization of the anteroposterior gradient, presence of any diffuse/focal slowing and amplitude of background activity. This procedure provided a classification of the EEG background into four categories, according to the clinical classification proposed in [23,24]: normal, mildly abnormal, moderately abnormal, and severely abnormal.

As in [18], the fraction of MCS patients with PCI_max_ > PCI* was computed to assess the sensitivity of PCI, whereas the PCI_max_ values observed in UWS patients were used to possibly detect the potentiality for consciousness in unresponsive patients. The Wilcoxon rank-sum test was applied to compare the PCI_max_ values between MCS+ and MCS− patients and to compare the CRS-R total scores between low-complexity (PCI_max_ ≤ PCI*) and high-complexity (PCI_max_ > PCI*) UWS patients. The percentages of high-complexity UWS patients obtained in this reproducibility study and in a previous reference study [18] were compared using Fisher’s exact test. The results obtained in each patient from behavioral clinical evaluation with the CRS-R as well as from neurophysiological assessment with PCI_max_ were integrated with qualitative evaluation of the EEG background. Specifically, the Kruskal–Wallis test was applied to compare the PCI_max_ values of MCS patients among mildly, moderately and severely abnormal EEG categories.

## 3. Results

### 3.1. Clinical Assessment

The 24 patients enrolled in this study (10 female; age range 19–55 years) suffered from DOC following a severe brain injury. Time since injury was ≥2 months for patients with anoxic etiology (*n* = 6) and ≥5 months for patients with other etiologies. Clinical diagnosis was UWS in 11 patients and MCS in 12 patients, including 4 MCS− and 8 MCS+ patients. One additional patient with traumatic etiology was diagnosed as emergence from MCS (EMCS). Traumatic etiology was prevalent among MCS patients (*n* = 7 as compared to *n* = 2 anoxic, *n* = 2 vascular and *n* = 1 other). In UWS patients, etiology was distributed as follows: *n* = 4 anoxic, *n* = 4 traumatic, *n* = 2 vascular, *n* = 1 other.

### 3.2. Neurophysiological Assessment

The experimental protocol applied in [18] recommended stimulating, whenever possible, the superior frontal and parietal cortex bilaterally, in each patient. Indeed, this previous work reported that about 71% of the patients were stimulated in 3 or more targets. This approach aims at maximizing the possibility of finding the best entry port for probing cortical reactivity and effective connectivity. In the present study, the spatial location and extent of cortical lesions and the progressive tiredness of patients during TMS-EEG recordings (as revealed either by a progressive difficulty to keep the eyes open or by an increasing motor agitation and facial muscle contraction) actually allowed stimulating 3 targets in 2 patients, 2 targets in 10 patients and 1 target in 12 patients. Thus, we acknowledge that the reduced number of stimulated targets might have led to underestimation in some of the individual PCI_max_ values.

PCI_max_ was higher than PCI* in the EMCS patient and in 11 out of 12 MCS patients (Figure 1a), yielding a sensitivity of 92% in the detection of minimal signs of consciousness. PCI_max_ did not significantly differ between MCS+ and MCS− patients (Wilcoxon rank-sum test, *p* = 0.2).

In the present UWS population, PCI allowed stratifying patients into two main subgroups (instead of three as in [18]), since a PCI_max_ = 0 was never found (Figure 1a). Most of the UWS patients (*n* = 7, 64%) constituted the low-complexity subgroup (PCI_max_ ≤ PCI*) and were characterized by a slow and stereotypical EEG responses to TMS, similar to the ones previously observed in healthy controls in the states of NREM sleep and anesthesia. Four UWS patients showed PCI_max_ > PCI*, thus representing the high-complexity subgroup: in these patients, TMS was able to elicit a rapidly changing and spatially differentiated cortical response, comparable to the one obtained in MCS patients and conscious controls. The CRS-R total scores were not significantly different between low-complexity and high-complexity UWS patients (Wilcoxon rank-sum test, *p* = 0.9).

Figure 1b shows examples of TMS-evoked potentials, voltage scalp topographies, and cortical maps of significant current density obtained in three representative patients: a high-complexity MCS patient and two UWS patients from the low-complexity and high-complexity subgroups.

Qualitative assessment of resting-state EEG was successfully performed in 22 patients (Figure 2a), because of insufficient data quality in 1 MCS and 1 UWS patient. The EMCS patient, 4 MCS patients (Figure 2a) and 1 high-complexity UWS patient (Figure 2b) were characterized by a mildly abnormal EEG background, further confirming the notion that a normal or mildly abnormal background allows the detection of conscious conditions with high specificity. A moderately abnormal pattern was observed in 5 MCS patients (including the one with low-complexity PCI_max_, Figure 2a) and 2 UWS patients (1 with high-complexity PCI_max_, Figure 2c). Thus, resting-state EEG activity was designated as either mildly or moderately abnormal in most MCS patients (*n* = 9, 82%); conversely, most UWS patients (*n* = 7, 70%, including 1 patient with high-complexity PCI_max_) showed a severely abnormal EEG background. Considering MCS patients, PCI_max_ was not significantly different among the three EEG categories (Kruskall–Wallis test, *p* = 0.1) (Figure 2c). Thus, in agreement with [18], EEG responses to TMS in MCS patients were characterized by high-complexity PCI values irrespective of the heterogeneous EEG background.

Outcome at 6 months was available for 7 patients: 4 high-complexity MCS patients recovered functional communication with the environment, 2 high-complexity UWS patients did not show any improvement and the high-complexity UWS patient with a mildly abnormal EEG background transitioned to MCS.

## 4. Discussion

In the present study, the sensitivity of PCI in the detection of minimal signs of consciousness was estimated at 92%; this result, although obtained on a relatively small patient population, is in line with the value of 94.7% reported in [18], thus confirming PCI as one of the best-performing neurophysiological markers across different patient populations. This result was obtained by applying the previously published [18] empirical cutoff PCI* validated on a large benchmark population. PCI successfully exploits the deterministic patterns of causal interactions among brain areas elicited by a direct and non-invasive perturbation of cortical neurons with TMS. Other quantitative, neurophysiological approaches based either on resting EEG or evoked potentials have shown lower sensitivity to detect MCS patients [25,26]. Nonetheless, they have been recently recommended to complement standard behavioral evaluation and to provide large-scale, first screening because of their broad availability and relative simplicity of application [16].

The lack of significant differences in PCI_max_ values between MCS+ and MCS− patients is also consistent with [18] and suggests that the complexity of brain responses to perturbation does not necessarily parallel the ability of generating complex types of behavior.

Most of the MCS patients (82%) showed either a moderately or a mildly abnormal background EEG, in agreement with the corresponding fraction (81.6%) reported in [18]. In addition, considering MCS patients, PCI_max_ was not significantly different among the three EEG categories, in agreement with [18]. Importantly, the heterogeneous EEG background observed in MCS patients (including 2 patients with a severely abnormal background) did not affect the complexity of EEG responses to TMS. Similarly, in our sample we observed one high-complexity UWS patient within each EEG category. Indeed, PCI_max_ clearly showed higher sensitivity than the qualitative evaluation of resting EEG in both studies. This result seems to confirm a relative advantage of perturbations and causal measures of complexity over observational measures of brain dynamics [17].

At the same time, in accordance with recent reviews on the role of neurophysiology in the evaluation of DOC patients [14,16], our results further confirm that a mildly abnormal EEG background represents a highly specific marker of a conscious condition (MCS or EMCS). Accordingly, the detection of this background in UWS patients should raise doubt about a correct clinical diagnosis (notably, in our sample, the only UWS patient with a mildly abnormal EEG also presented high complexity to TMS/EEG evaluation and subsequently transitioned to MCS).

UWS patients were stratified by PCI_max_ into a low-complexity subgroup of 7 patients (64%) and a high-complexity subgroup of 4 patients (36%). In comparison, an analogous subdivision performed in [18] yielded a no-response subgroup of 13 patients (30%), a low-complexity subgroup of 21 patients (49%), and a high-complexity subgroup of 9 patients (21%). Considering that, in [18], anoxic etiology was prevalent among UWS patients with PCI_max_ = 0, the absence of the no-response subgroup in the present study might be related to the small number of post-anoxic UWS patients in this sample (*n* = 4). The percentages of high-complexity patients among all the UWS patients are not significantly different between the two studies (Fisher’s exact test, *p* = 0.4). The absence of significant difference in the CRS-R total scores between the low-complexity and high-complexity UWS subgroups is also consistent with the results of [18]. These findings suggest that PCI_max_ may provide novel information about UWS patients unavailable from the analysis of their behavior. Specifically, the similarity of the TMS-evoked potentials obtained in high-complexity UWS patients to those observed in conscious controls and MCS patients may indicate a covert capacity for conscious information processing.

All the MCS patients for whom the outcome is known and who recovered functional communication within 6 months (*n* = 4) showed high-complexity TMS-evoked potentials (with PCI_max_ > PCI*). Among the UWS patients, outcome data were available for two high-complexity patients, who did not show any improvement, as well as for the high-complexity UWS patient with a mildly abnormal EEG background, who transitioned to MCS. Although, in this sample, the rate of transitioning to MCS among the high-complexity UWS patients with known outcome (1 out of 3) was smaller than in [18] (6 out of 8), the difference was not statistically significant (Fisher’s exact test, *p* = 0.5). The small sample size, the variable onset time from the brain injury and the incomplete information about the outcome for all patients do not allow proper testing of the potential of PCI_max_ in recovery prediction.

Similarly to previous observations in healthy subjects during deep sleep and anesthesia [18,27], in the present study, low-complexity UWS patients showed a slow and stereotypical EEG response to TMS, associated with a significant suppression of high-frequency activity (Appendix A). This result replicates a previous study by [28], in which the loss of complexity after severe brain injury was related to a pathological propensity of cortical neurons to enter a state of inactivity (OFF-period) after receiving an input. The present study contributes to the reliable identification of this mechanism and thus encourages the development of effective therapeutic options [5] targeting intrinsic neuronal properties for a concurrent restoration of brain complexity and recovery of consciousness.

## 5. Conclusions

This study has successfully replicated the performance of PCI in discriminating between UWS and MCS patients, previously reported in [18]. This result further motivates the application of TMS-EEG in the workflow of DOC evaluation, especially for unresponsive patients with either a moderately or a severely abnormal EEG background [16]. Since high-complexity PCI values have always been associated with consciousness [18], also in the present study it is parsimonious to assume that a capacity for consciousness not apparent from behavior may exist in the UWS patients with high-complexity PCI. Thus, this study further suggests that high-complexity UWS patients might specifically benefit from active paradigms or brain–machine interface approaches aimed at ultimately establishing communication with the external environment [29,30,31].

## Figures and Tables

**Figure 1 brainsci-10-00917-f001:**
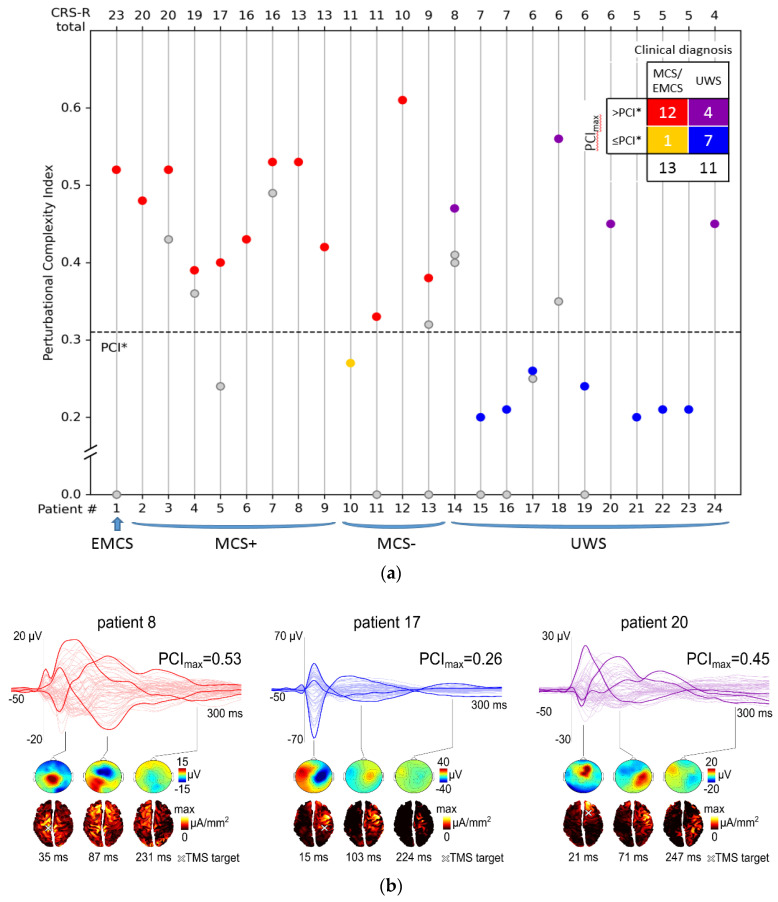
(**a**) Distribution of perturbational complexity index (PCI) values computed from patients with disorders of consciousness: unresponsive wakefulness syndrome (UWS), minimally conscious state (MCS+/−, according to [21]), emergence from MCS (EMCS). The dashed horizontal line corresponds to the empirical cutoff (PCI*) previously validated in a large benchmark population to discriminate between unconsciousness and consciousness [18]. Multiple PCI values computed in single patients are aligned along vertical lines. Within each diagnostic category, patients are sorted by decreasing Coma Recovery Scale-Revised (CRS-R) total score. Gray circles represent submaximal PCI values, whereas individual maximum PCI values (PCI_max_) are depicted by colored circles. The contingency table displayed in the upper right corner summarizes how many patients for each diagnostic category show a PCI_max_ value either > PCI* or ≤ PCI*. (**b**) Top row shows the average transcranial magnetic stimulation (TMS)-evoked potentials (all channels superimposed, with 3 illustrative channels highlighted in bold) together with the PCI_max_ values for three representative patients: a high-complexity minimally conscious state (MCS) patient (red traces), a low-complexity unresponsive wakefulness syndrome (UWS) patient (blue traces) and a high-complexity UWS patient (purple traces). For each patient, voltage scalp topographies (second row) and significant current density cortical maps (third row) are shown at three selected time points. A white cross on the cortical map indicates the stimulation site.

**Figure 2 brainsci-10-00917-f002:**
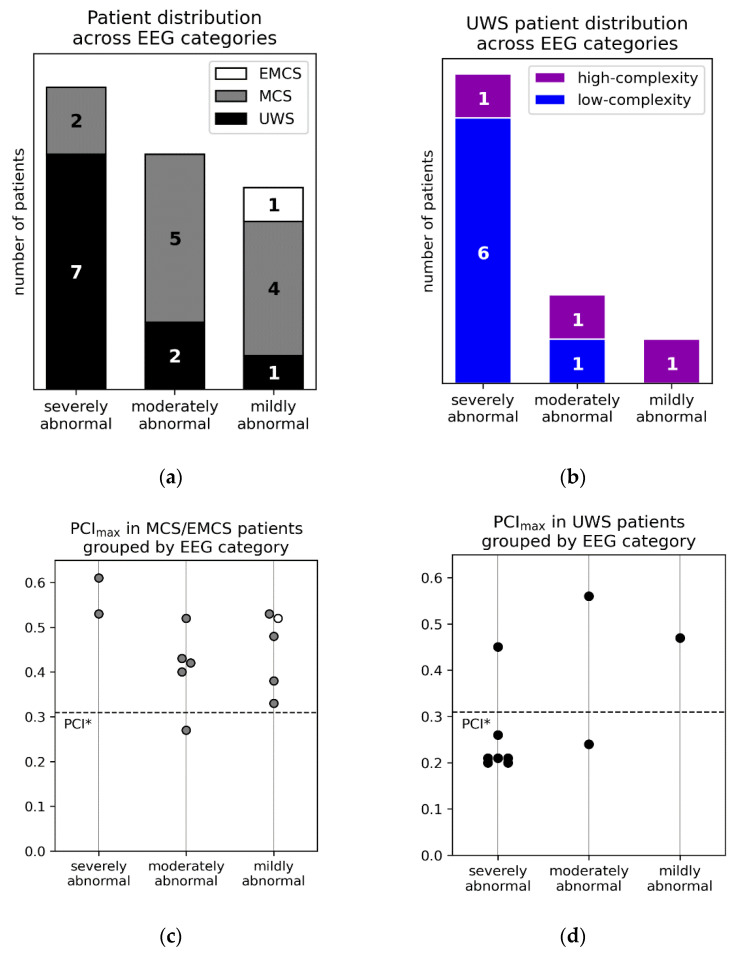
(**a**) Distribution of all patients (MCS = minimally conscious state, EMCS = emergence from MCS, UWS = unresponsive wakefulness syndrome) across background EEG categories, i.e., severely, moderately, and mildly abnormal. (**b**) Distribution of low-complexity (i.e., with PCI_max_ ≤ PCI*) and high-complexity (i.e., with PCI_max_ > PCI*) UWS patients across background EEG categories. Individual PCI_max_ values in MCS (gray circles) and EMCS (white circle) patients (**c**) and in UWS (black circles) patients (**d**) grouped by background EEG category. The dashed horizontal line corresponds to the empirical cutoff PCI* previously validated in [18].

**Table 1 brainsci-10-00917-t001:** Clinical and neurophysiological characteristics of disorders of consciousness (DOC) patients. The Coma Recovery Scale Revised (CRS-R) evaluation corresponds to the best diagnosis out of five assessments (for multiple evaluations with this diagnosis, the one with the highest score is reported).

					CRS-R												
Patient Number	Age [Years]/Gender	Time Since Injury [Months]	Etiology	Clinical Diagnosis	Auditory	Visual	Motor	Verbal	Comm	Arousal	TOT	Outcome	Stimulated Sites Number	PCI_max_ Brain Side	PCI_max_ Brain Area	PCI_max_	EEG Category ^‡^
1	20/m	24	T	EMCS	4	5	6	3	2	3	23	**conscious**	2	L	F	0.52	Mi
2	20/m	15	T	MCS+	4	5	5	2	1	3	20	**conscious**	1	L	F	0.48	Mi
3	29/m	17	T	MCS+	4	4	5	3	1	3	20	unknown	3	R	F	0.52	Mo
4	21/f	12	T	MCS+	4	5	5	2	0	3	19	**conscious**	2	L	F	0.39	n/a
5	32/m	32	T	MCS+	4	4	5	1	1	2	17	unknown	2	R	P	0.40	Mo
6	31/f	5	hV	MCS+	4	4	5	1	0	2	16	unknown	1	R	P	0.43	Mo
7	29/m	56	T	MCS+	2	4	5	2	1	2	16	**conscious**	2	L	F	0.53	Mi
8	55/m	4	A	MCS+	2	2	4	2	1	2	13	unknown	1	L	P	0.53	Se
9	43/f	7	other ^¥^	MCS+	3	3	3	1	1	2	13	**conscious**	1	L	P	0.42	Mo
10	44/m	9	T	MCS−	2	2	3	2	0	2	11	unknown	1	L	F	0.27	Mo
11	26/m	5	T	MCS−	2	3	3	1	0	2	11	unknown	2	R	P	0.33	Mi
12	47/m	9	A	MCS−	2	3	2	1	0	2	10	unknown	1	R	P	0.61	Se
13	48/m	13	hV	MCS−	2	2	2	1	0	2	9	unknown	1	R	F	0.38	Mi
14	24/f	22	T	UWS	1	1	2	2	0	2	8	**MCS**	3	L	F	0.47	Mi
15	25/f	12	T	UWS	1	1	2	1	0	2	7	unknown	2	R	F	0.20	Se
16	47/m	3	A	UWS	2	0	2	1	0	2	7	unknown	2	R	F	0.21	Se
17	49/f	7	other ^§^	UWS	1	0	2	1	0	2	6	unknown	2	R	F	0.26	Se
18	34/f	9	A	UWS	1	0	2	1	0	2	6	UWS	2	L	F	0.56	Mo
19	47/m	30	iV	UWS	1	0	2	1	0	2	6	unknown	2	R	P	0.24	Mo
20	19/f	6	T	UWS	1	0	2	1	0	2	6	UWS	1	R	F	0.45	Se
21	47/f	26	hV	UWS	0	0	2	1	0	2	5	unknown	1	L	P	0.20	Se
22	22/m	12	T	UWS	1	0	2	1	0	1	5	unknown	1	R	F	0.21	Se
23	51/m	10	A	UWS	0	1	1	1	0	2	5	unknown	1	L	P	0.21	Se
24	27/f	2	A	UWS	0	0	1	1	0	2	4	unknown	1	L	P	0.45	n/a

m = male, f = female; A = anoxic, T = traumatic, iV = vascular ischemic; hV = vascular hemorrhagic; ^§^ = infection of the central nervous system, ^¥^ = hyperglycemia; Comm = communication CRS-R score; TOT = total CRS-R score; L = left, R = right; F = superior frontal cortex, P = superior parietal cortex; ^‡^ = EEG category based on [23,24], Mi = mildly abnormal, Mo = moderately abnormal, Se = severely abnormal. Known outcomes with a change of consciousness state are shown in bold.

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
