# Peer review of "Detecting the Potential for Consciousness in Unresponsive Patients Using the Perturbational Complexity Index"

_brainsci, 2020, doi:10.3390/brainsci10120917_

Round 1
Reviewer 1 Report
I have no particular concern with this study.
However, it does not contain any novel result. It is confirmatory regarding the utility of the Perturbational Complexity Index in the diagnosis of patients with disorders of consciousness, in a relatively small sample of patients.
Reviewer 2 Report
Manuscript ID: brainsci-964833
Title: Detecting the potential for consciousness in unresponsive patients using the perturbational complexity index
Journal: Brain Sciences
SUMMARY OF WORK
In this paper, the authors test the ability for the perturbational complexity index (PCI) to discriminate between patients at different levels of disorders of consciousness (DOC). Replicating the methods and analysis from Casarotto et al (2016), this study uses transcranial magnetic stimulation (TMS) to perturb the brains of patients in DOC and measures the complexity of the response through EEG. The authors find that the PCI values obtained from 11 of the 12 individuals in minimally conscious state (MCS) fall above the previously-established critical threshold of PCI = 0.31, and that most UWS patients showed a low-complexity response to TMS. A patient’s PCI was a better indicator of their level of consciousness than clinical assessment of their resting-state EEG.
General Comments
While this is in large part a replication study, the value of this contribution should not be underestimated. As the authors point out, it is important to increase the evidence associated with the diagnostic accuracy of the PCI with a novel set of patients, a different clinical context, and a different set of equipment and experimenters. The fact that this study was able to replicate the findings of the Casarooto et al (2016) paper strengthens the evidence for use of PCI in the assessment of DOC patients, and contributes to a stronger recommendation of this metric in future guidelines with this population.
Specific Comments
- The introduction is excellent. Well-referenced, and an accurate synthesis of the current literature and state-of-the-art in DOC research.
- Line 86 – please change “stable clinical condition from at least two weeks” to “stable clinical condition for at least two weeks”
- The methods were clear and complete.
- Line 165 – please change “to find” to “of finding”
- Line 166 – how did the authors assess progressive tiredness? The patient’s inability to maintain wakefulness, in spite of deep pressure stimulation?
- Line 182 – I believe that this should be Figure 1b instead of 2b.
- Figure 1a. Please move the label for the x-axis below the axis. This figure has a lot of blank space – it would be more striking to begin the y axis at 0.1 instead of 0, and end the y axis at 0.7 to maximize the space with data.
- In the discussion, the authors mention that PCImax shows higher sensitivity than the qualitative evaluation of resting EEG. Many more sophisticated EEG techniques exist for evaluating disorders of consciousness - I would suggest adding a sentence or two to the discussion acknowledging this body of literature.
Reviewer 3 Report
The authors assessed the reliability of the perturbational complexity index (PCI), which is a measure of complexity of EEG response to TMS, in a small sample of patients with disorders of consciousness, including 12 MCS, 1EMCS and 11 UWS patients. They found that PCI can be useful for distinguishing different states of consciousness and disclosing covert capacity of consciousness.
The findings of the present study are in line with previous results, confirming the usefulness of PCI and TMS-EEG as advanced tools in the clinical practice for the diagnosis of disorders of consciousness.
I found the article interesting and of interest for the readers of the journal. However, I have some concerns that the authors should address.
- The clinical assessment (CRS-R) is only mentioned and never defined. A brief description of this assessment in the methods would help a wider range of readers.
- Were the patients under any kind of medication at the time EEG and TMS-EEG data collection? Could drugs have affected the PCI values?
- Lines 143-145, could the Authors better specify how they classified the EEG background into four categories?
- Can the Authors provide explanation for segmenting EEG in 1.6sec-long epochs around the TMS pulse?
- The sample included is heterogeneous for etiology; did the Authors test statistically whether PCI values differ across different etiologies (e.g traumatic vs vascular)?
- Could the Authors add a “statistical analysis” paragraph in the methods section? They mention a number of statistical tests both in the results and discussion sections, without mentioning them in the methods section. Despite this is not strictly necessary, I think it may make clearer the statistical design of the paper.
- I would suggest moving supplementary table 1 in the main text, as it provides the readers with useful information on clinical, behavioral and neurophysiological characteristics.
The present study further validated TMS-EEG and PCI for detecting minimal signs of consciousness in an independent small sample of DOC patients. I hope the Authors take into consideration reviewer’s comment which may help in improving the readability of the work.
Reviewer 4 Report
Sinitsyn et al. measured perturbational complexity index (PCI) in prolonged and chronic DOC patients. The authors replicated a previous study by Casarotto et al. (2016) and supported the potential utilization of PCI method in assisting clinical assessment and evaluation for DOC patients. Although the present work did not show any conceptual or empirical advances beyond the previous study (Casarotto et al., 2016), replicating the main results derived from a promising TMS-EEG method in the complicated and challenging clinical settings is of importance. In other words, studies supporting the reproducibility of preclinical neuroimaging methods with strong clinical relevance should be encouraged.
I have a few comments to share:
- Unlike the samples in Casarotto et al. (2016), in this study, there is no control group such participants during wakefulness or anesthesia serving as the ground truth for consciousness vs. unconsciousness. Therefore, it is hard to assert the genuine sensitivity or reliability by only testing the MCS and UWS patients, unless the authors assume the behavioral assessment by CRS-R could offer such ground truth. However, the clinical assessment based on behavior response can lead to erroneous conclusions in up to 40% of patients. I wonder if the author could constrain their interpretations and discuss the limitations in this regard.
- In the abstract, the authors stated that “…sample of 24 severely brain-injured patients, including 12 minimally conscious state (MCS) and 11 unresponsive wakefulness syndrome (UWS) patients.” This leads some confusion as 12+11=23. After reading the methods part, I realized that there was one additional patient with traumatic etiology diagnosed as emergence from MCS (EMCS). I would suggest adding this info in the abstract.
- Unfortunately, the outcome at 6 months was only available for 7 patients and the results are mixed. For example, 2 high-complexity UWS patients did not show any improvement and the high-complexity UWS patient transitioned to MCS. Therefore, the prognosis value of PCI is entirely unclear. I wonder if there will be further updates on the outcomes of other UWS patients. This will be clinically informative, because we would otherwise never know if PCI could inform a better therapeutic strategy at the time of study.
Reviewer 5 Report
The authors of the article tested the credibility of the perturbational complexity index in 24 severely brain injured patients. The basic sections (introduction, materials and methods, results, conclusion, literature cited, etc.) are adequate. The author wrote clear subheadings and they were relevant to the text as they clarified the sections of the text. The material order is easy to follow. The author's writing style was clear.
In the introduction the authors explain the reasons for conducting their research as well as its importance, however in some parts the explanations are not sufficient. For example the authors write “The clinical characterization of DOC is important for prognosis and for personalizing the rehabilitation treatment» but do not explain their point of view further.
In the Experimental protocol subsection the authors often invoke the experimental procedures of another research, it is recommended to describe the procedures a little further in order for the article to be more self-sufficient.
In the discussion section the authors state that “n the present study, the sensitivity of PCI in detecting minimal signs of consciousness was 235 estimated at 92%; although this result may be affected by the relatively small sample size, this value is considerably higher in comparison with other EEG-based approaches”, but give no explanation as to the possible reasons why such differences exist.
Round 2
Reviewer 1 Report
no further comments
Author Response
We thank the Reviewer again for the assessment of the manuscript.